

# Efficacy of an improved method to screen semiochemicals of insect

Chongyang Li, Jianmin Cao, Xiufang Wang, Pengjun Xu, Xinwei Wang and Guangwei Ren

Qingdao Special Crops Research Center, Chinese Academy of Agricultural Sciences, Qingdao, China

## ABSTRACT

**Background:** A combination of gas chromatography-electroantennographic detection (GC-EAD) and gas chromatography-mass spectrometry (GC-MS) is typically used to screen active compounds that play a role in the regulation of insect behavior. This method uses two kinds of gas chromatography (GC) equipment and needs to compare compounds between the two chromatograms, and it is tedious and costly. To improve detection efficiency, as well as reduce costs and the rate of missed detection, we designed a system connecting gas chromatography (GC), electroantennography (EAG), and mass spectrometry (MS), with MS used instead of the flame ionization detector (FID) as the GC-EAD detector. To verify the feasibility of the improved method, we compare two methods—GC-EAG-MS and GC-EAD—through a series of experiments. Some researchers made similar improvements, but these were not compared with GC-EAD, and their method needed to be improved in the synchronization and split ratio. Our method has been optimized and improved in these aspects.

**Methods:** *Helicoverpa armigera* was the test organism; the improved method and conventional method were used to detect known and unknown compounds, as well as screen out active compounds that could generate responses in *H. armigera* antennae.

**Results:** Screening known single compounds using the two methods, the active compound benzaldehyde was detected in all seven concentrations of solution. By using the two methods, the five same active compounds of *Helicoverpa armigera* were detected in high concentration solution of the mixed compounds (100 mg L$^{-1}$, 50 mg L$^{-1}$); the four same active compounds were detected at 20 mg L$^{-1}$ concentration; two identical same compounds were detected in low concentration solution (concentrations of 10 mg L$^{-1}$ and below). By using the two methods, six identical active compounds of *Helicoverpa armigera* were detected in unknown compounds.

**Conclusion:** The improved method was consistent with the conventional method in terms of accuracy and sensitivity. However, compared with the traditional methods, Gas chromatography-electroantennographic-mass spectrometry (GC-EAG-MS) saved the cost of GC and FID equipment, thereby greatly lowering the experimental cost. In the experiment, GC-EAG-MS combined the two experimental operations of screening active substances by GC-EAD and identifying active substances by GC-MS into one, which not only reduced the experimental steps, but also avoided the false positive caused by the comparison of the two chromatograms, and it greatly

Corresponding author
Guangwei Ren,
renguangwei@caas.cn

reduced the difficulty level of the overall experimental analysis. GC-EAG-MS is more convenient, efficient, economical, and practical, and could confidently replace traditional methods. With further optimization, it could be widely applied in the study of plant and insect chemical ecology.

# INTRODUCTION

Plant volatile compounds and insect pheromones are important substances that convey information to insects and play an irreplaceable role in their foraging, feeding, mating and reproductive activities. Electroantennography (EAG) (*Schneider, 1957*) and gas chromatography-electroantennographic detection (GC-EAD) (*Moorhouse et al., 1969*; *Arn, Städler & Raucher, 1975*) are effective methods for studying the recognition and perception of these substances in insects, with GC-EAD being widely used to screen chemical mixtures of plant volatile compounds and insect pheromones (*Huang et al., 2014*; *Johnson et al., 2020*; *Munro et al., 2020*; *Pawlowski, Sweeney & Hillier, 2020*).

Currently, the following methods are used to study collected chemical mixtures (*Huang et al., 2014*; *Milonas, Anastasaki & Partsinevelos, 2019*; *Munro et al., 2020*; *Ortiz-Carreon et al., 2019*): Active components that play a role in insect behavior regulation are screened using GC-EAD, and the whole samples are analyzed using gas chromatography-mass spectrometry (GC-MS). EAD active compounds are identified by GC-MS analysis according to their mass spectra and retention times, in comparison with synthetic or authentic standards. Although this method can identify the composition of information material quickly and accurately, it presented limitations in practical applications. For example, (1) GC-EAD and GC-MS are two separate systems, often using different gas chromatography systems. Due to differences in type, carrier gas, and columns between these two gas chromatography systems, the peak shape, retention time and quantity of the analyzed compounds were varied as well. We have to spend a lot of energy comparing the corresponding compounds in the two chromatograms to prevent incorrect identification results, resulting in a more time-consuming experiment. (2) The application of this method required the combination of GC-EAD and GC-MS, which was tedious and costly. (3) If the sample quantity was too small to support the second injection, it was impossible to carry out analysis and screening. The present study aimed to improve the GC-EAD technology by adding a capillary flow purged splitter in the gas chromatography (GC) structure. The GC column outlet was split between EAG and MS using a capillary flow purged splitter with makeup gas. The flame ionization detector (FID) was replaced by mass spectrometry (MS). This improved method combined the two experimental operations of GC-EAD and GC-MS in one system, which reduced the cost of one GC equipment and one FID equipment, and avoided comparison of different chromatograms. It reduced the time and operation, and improved the efficiency.

The GC-EAD method is widely recognized by researchers. We improved this method and carried out comparative experiments between the new method and the original method (GC-EAD) to verify whether the improvement is accurate and feasible. *Weissbecker, Holighaus & Schütz (2004)* made similar improvements, and this method was used by *Paczkowski et al. (2013)*. However they did not perform any comparative experiments with the original method (GC-EAD). Therefore, our study is an important supplement to the work of *Weissbecker, Holighaus & Schütz (2004)*. The method described by *Weissbecker, Holighaus & Schütz (2004)* needs to be improved in the synchronization and split ratio, and our study has improved these aspects.

To verify the efficacy of the improved method, we screened active substances in known and unknown detected compounds and compared them with data obtained by conventional methods. In this experiment, the single known compound was the artificially configured benzaldehyde solution. The mixed compounds were a mixture of cis-3-hexen-1-ol, myrcene, linalool, methyl salicylate, and trans-β-caryophyllene. The tested insects were *Helicoverpa armigera*, and the unknown compounds were the headspace volatiles of *H. armigera* lures.

In this study, we improved a method of the combined use of GC, EAG and MS. Then we verified the feasibility of this method through a series of experiments. The results showed that the improved method could obtain same results as the conventional method (GC-EAD) under the same conditions, and the improved method was more convenient and efficient. This paper could provide a reference method for the study of chemical ecology of plants and insects.

## MATERIALS & METHODS

### Insects

*H. armigera* purchased from Henan Jiyuan Baiyun Industrial Co., Ltd. (China) were fed for several generations under controlled conditions at a 14-h light: 10-h dark photoperiod, 26 ± 1 °C, and 55–65% relative humidity (*Sun, Huang & Wang, 2012*). Larvae were fed an artificial diet. Pupae were sexed and separated accordingly in cages.

### Chemicals

Benzaldehyde (98.5%) was purchased from Sinopharm Chemical Reagent Co., Ltd., (Shanghai, China). Myrcene (90%), trans-β-caryophyllene (80%), cis-3-hexen-1-ol (98%), linalool (98%), eugenol (99%), benzyl alcohol (99.5%), 2-ethyl-p-xylene (98%), α-terpineol (98%) and cis-11-hexadecenal (95%) were purchased from Shanghai Macklin Biochemical Co. Ltd. (China), and methyl salicylate (99.5%) was purchased from Tianjin Beilian Fine Chemicals Development Co., Ltd. (China). Hexane was purchased from Merck KGaA (Darmstadt, Germany).

To facilitate the experiment, the compounds we selected were common floral substances. It was difficult to generate an antennal reaction when the concentration of most compounds was lower than 1 mg L$^{-1}$; we therefore set the minimum test concentration of the artificial solution to 1 mg L$^{-1}$.

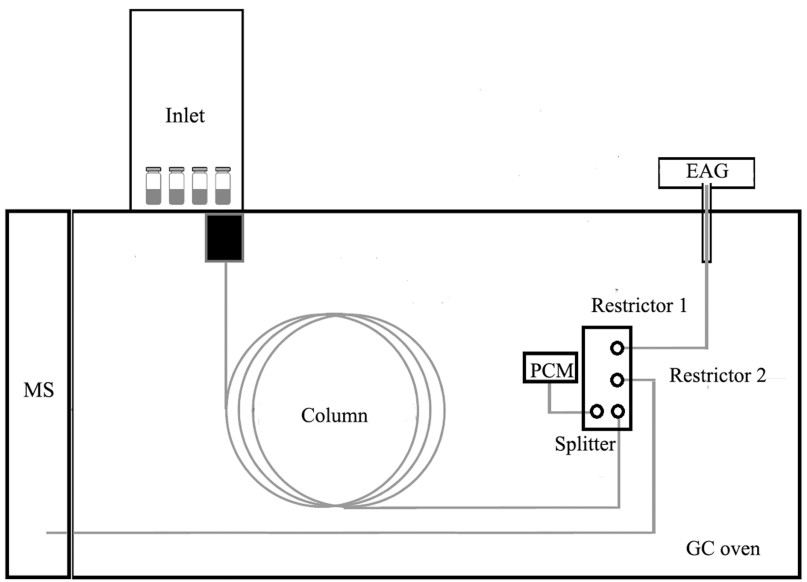

**Figure 1 The composition of GC-EAG-MS.**

The benzaldehyde solution was configured with hexane as solvent and was divided into seven concentration gradients, i.e., 100, 50, 20, 10, 5, 2 and 1 mg $L^{-1}$.

A mixture of cis-3-hexen-1-ol, myrcene, linalool, methyl salicylate, and trans-β-caryophyllene was prepared using hexane as the solvent, at a concentration gradient similar to the above.

*Helicoverpa armigera* lures were purchased from Pherobio Technology Co., Ltd. (China).

## Collection of volatiles

Headspace sampling was used to collect volatiles from *Helicoverpa armigera* lures (*Sun, Huang & Wang, 2012*; *Wei, Zhu & Kang, 2006*) enclosed in a 5-L gas sampling bag. Volatiles were extracted from the bag using an air sampler at a rate of 200 mL $min^{-1}$ and trapped in a glass tube (8 cm long, with an inner diameter of 6 mm) containing 60 mg of 60/80 mesh Tenax TA adsorbent (Alltech Assoc., Deerfield, IL, USA) with frits (Agilent Technologies, Santa Clara, CA, USA) at both ends. The collected air was then passed through a freshly activated charcoal filter for purification and re-flowed into the gas sampling bag, to form a cycle. Each collection procedure lasted for 12 h and was replicated five times. Volatiles were desorbed with 600 μL hexane and stored at −20 °C for subsequent use.

## Gas chromatography-electroantennographic-mass spectrometry analyses

Gas chromatography, electroantennography, and mass spectrometry were used together, with the following specific improvement and application methods (Fig. 1).

A capillary flow purged splitter with makeup gas (G3180B; Agilent, Santa Clara, CA, USA) and pneumatic control module (PCM, G3471A; Agilent, Santa Clara, CA, USA)

were installed in the GC. This system could split the effluent of the GC column in a proper proportion and ensure that the MS was under high vacuum and the EAG was at atmospheric pressure. GC (7890B; Agilent, Santa Clara, CA, USA) was equipped with an HP-5ms Ultra Inert column (30 m × 250 μm ID and a film thickness of 0.25 μm, Agilent Technologies, Santa Clara, CA, USA). The system needed to meet the following requirements: the effluent of the GC column was split in a ratio of 1 (MS) to 3 (EAG), and the two shunt substances arrive at their respective detectors at the same time. Therefore the specifications of the columns equipped with MS and EAG were different. We used the 2-Way Effluent Splitter Calculator (with Makeup) (Agilent Technologies, Santa Clara, CA, USA) to calculate the column specifications installed in MS and EAG. MS (7010B; Agilent, Santa Clara, CA, USA) was equipped with a DB5MS column (2.75 m × 150 μm ID, Agilent Technologies, Santa Clara, CA, USA), and EAG (Syntech, Battaramulla, Sri Lanka) was equipped with a DB5MS column (1 m × 200 μm ID, Agilent Technologies, Santa Clara, CA, USA). This setting could meet the above requirements. The carrier gas was high-purity helium (2.25 mL min$^{-1}$). The oven temperature was programmed as follows: initially set at 50 °C for 2 min, increased to 260 °C at a rate of 10 °C min$^{-1}$, and maintained for 3 min.

EAG specifications: An antenna of one- to two-day-old adults was cut at the base of the flagellum, and the tip of the terminal segment was removed. Using an electrode gel, the excised antenna was mounted on a microelectrode connected to a micromanipulator (MP-15; Syntech, Battaramulla, Sri Lanka). The stimulus controller unit (CS-55; Syntech, Battaramulla, Sri Lanka) provided a stable airflow by maintaining a flow rate of 20–30 mL min$^{-1}$. The effluent conditioner assembly (TC-02; Syntech, Battaramulla, Sri Lanka) was set to a temperature of 250 °C, to prevent the sample from condensing. Signals were amplified with a USB acquisition controller (IDAC-2; Syntech, Battaramulla, Sri Lanka) and transferred to a computer. Data collection and processing were performed using the GC-EAD 2010 software (Syntech, Battaramulla, Sri Lanka) (*Sun, Huang & Wang, 2012*; *Zhao, Yan & Wang, 2006*). The compounds that flowed to EAG were to the antenna mixed with a charcoal-filtered and humidified air stream. Mounted antennae were placed in the extended end of a glass tube (inner diameter of 8 mm). Both the benzaldehyde solution and mixed solution were detected in female antennae, while volatiles of *H. armigera* lures were detected in that of males. Ten EAG recordings were obtained. A response was considered genuine if it was present in at least seven out of the ten replicates collected (*Noushini et al., 2019*).

MS specifications: the mass spectrometer was operated with the transfer line set at 250 °C, quadrupole at 150 °C, and ion source at 230 °C. Electron impact ionization was employed, with an electron energy of 70 eV. The mass range was set at 25–300 m/z. The compounds that flowed to MS were identified by comparing their mass spectra with that of the NIST library (Agilent Technologies, Santa Clara, CA, USA), and confirmed with authentic reference compounds.

During practical applications, the reaction peak start time of the active compound in EAG was consistent with that detected in MS. Therefore, we only needed to record the starting time of the EAG peak of the active compound, and found the corresponding

compound at this time in mass spectrum. The corresponding compound was this active compound, and identified it by MS.

The difference between the starting times of the peak recorded by the two software programs was approximately 0.01–0.03 min; this error margin was caused by manual operation. This system could be used not only to screen and identify active compounds, but also eliminate the use for Capillary Flow Purged Splitter and be used as GC-MS alone.

A 2 µL sample was introduced into the GC column by using GC-EAG-MS. Owing to limitations in the experimental set up, GC-EAD was used in a ratio of 1:1, which was different from GC-EAG-MS (1:3). To compare GC-EAD with GC-EAG-MS, we used an injection volume of 3 µL GC-EAD, to ensure that the volume of sample reaching the antennae was the same for both methods. This comparison was used to test the accuracy of the experimental results obtained using GC-EAG-MS. To test for the sensitivity of GC-EAG-MS, we used an injection volume of 1 µL GC-EAD, to ensure that the volume of sample reaching the detector (MS, FID) was the same for both methods.

## Gas chromatography and electroantennographic detection analyses

EAG specifications: data collection and processing were performed using the GC-EAD 2014v1.2.5 software (Syntech, Battaramulla, Sri Lanka), and the other specifications were set up as described for the GC-EAG-MS system above.

Linked GC-EAD analyses were performed using a Shimadzu 2030 GC instrument equipped with an RTX-5 column (30 m × 250 µm ID, and a film thickness of 0.25 µm), and nitrogen (1.4 mL min$^{-1}$) as carrier gas. Each sample was injected into the GC column in a splitless mode. Injection was performed at 220 °C and FID at 280 °C. Temperature programs were as per the GC-EAG-MS system described above. The effluent was partly to the FID and partly to the antenna mixed with a charcoal-filtered and humidified air stream. Mounted antennae were placed in the extended end of a glass tube (inner diameter of 8 mm).

## Gas chromatography and mass spectrometry analyses

The compounds were analyzed using a GC-MS system in traditional methods. GC (7890B; Agilent, Santa Clara, CA, USA) was equipped with an HP-5ms Ultra Inert column (30 m × 250 µm ID, film thickness was 0.25 µm, Agilent Technologies, Santa Clara, CA, USA). MS (7010B; Agilent, Santa Clara, CA, USA) was equipped with a DB5MS column (2.75 m × 150 µm ID, Agilent Technologies, Santa Clara, CA, USA). The carrier gas was high-purity helium (2.25 mL min$^{-1}$). Temperature programs were as per the GC-EAG-MS system described above. MS specifications were set up as described for the GC-EAG-MS system above.

# RESULTS

## Screening of known single compounds using two methods
### GC-EAG-MS analyses and identification of benzaldehyde solution
Analyzed EAG data, one compound elicited consistent responses in female *H. armigera* antennae (Fig. 2A). Recorded the starting time of the EAG peak of the active compound,

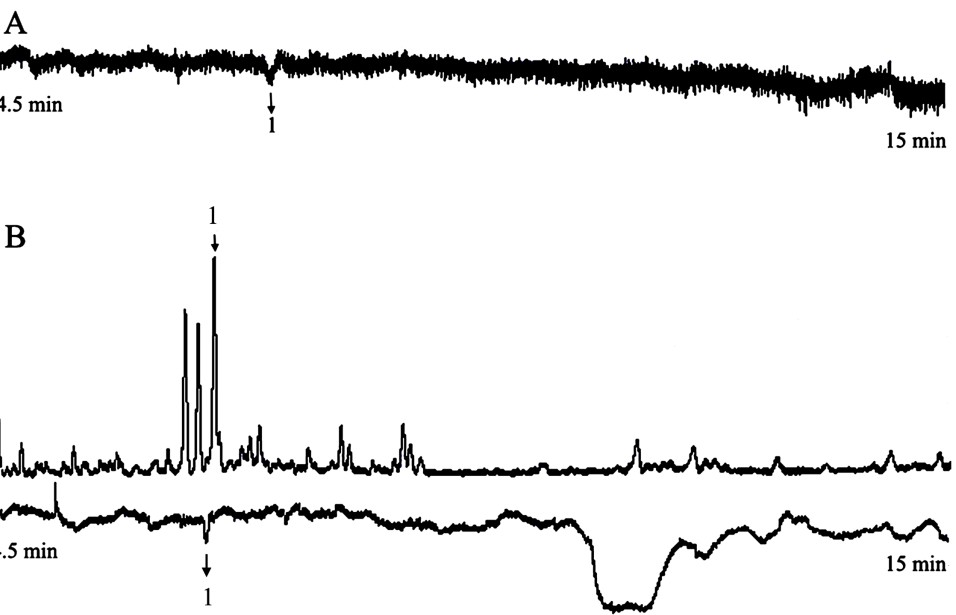

**Figure 2** The responses of female *Helicoverpa armigera* to benzaldehyde solution. (A) GC-EAG-MS, (B) GC-EAD. The active compound: (1) benzaldehyde.

and found the corresponding compound at this time in mass spectrum. It was identified as benzaldehyde by MS, according to their mass spectra and retention time, in comparison with authentic standards. All seven concentrations of the solution produced similar results.

Because this method only needed to determine the compound according to the start time of the EAG reaction peak, and this method does not use FID. Therefore, we only attached the corresponding EAG diagram in the paper (similarly hereinafter).

### GC-EAD and GC-MS analyses and identification of benzaldehyde solution

The injection volume for GC-EAD was 3 µL. In GC analyses with FID, one compound elicited responses in female *H. armigera* antennae (Fig. 2B). The injection volume for GC-MS was 3 µL. Analyzed and compared the corresponding compounds in the two chromatograms, the EAD active compound was identified as benzaldehyde using GC-MS analysis according to their mass spectra and retention times, in comparison with authentic standards. All seven concentrations of the solution produced the same result.

The injection volume for GC-EAD was 1 µL. The target compound benzaldehyde was detected in all seven concentrations of solution.

## Screening of mixed known compounds using two methods
### GC-EAG-MS analyses and identification of the mixed solution

Analyzed EAG data, five compounds consistent elicited responses in female *H. armigera* antennae at 100 mg L$^{-1}$ and 50 mg L$^{-1}$ concentrations (Fig. 3A). Recorded the starting time of the EAG peak of the active compounds, and found the corresponding compounds at these times in mass spectrum. These were identified by MS, according to their mass spectra and retention times, in comparison with authentic standards. They were cis-3-

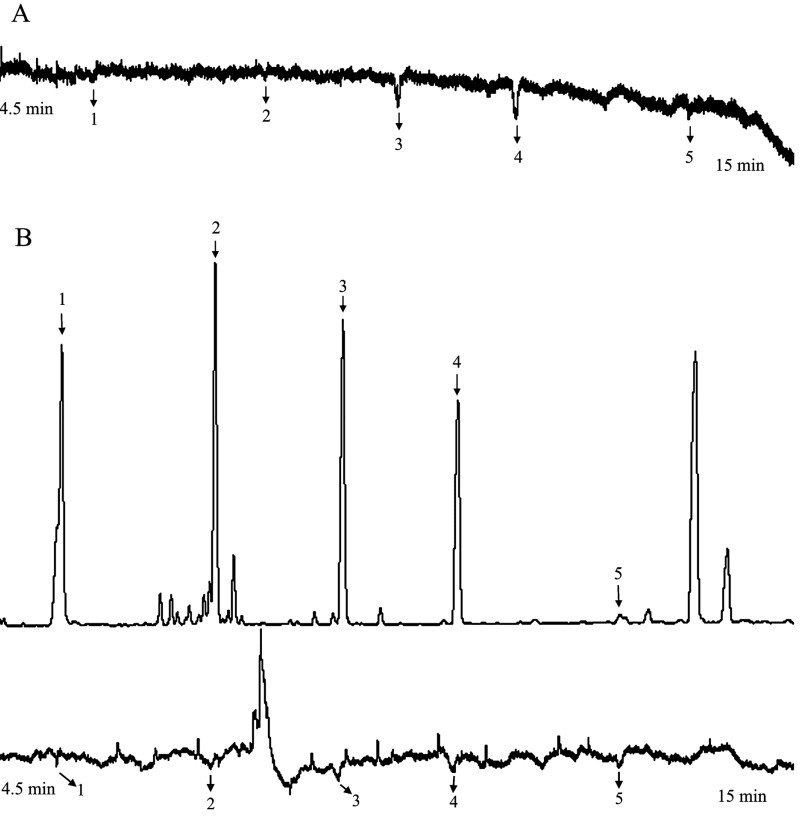

**Figure 3 The responses of female *Helicoverpa armigera* to the mixed solution (50 mg L$^{-1}$).** (A) GC-EAG-MS, (B) GC-EAD. The active compound: (1) cis-3-hexen-1-ol; (2) myrcene; (3) linalool; (4) methyl salicylate; (5) eugenol.                                         

hexen-1-ol, myrcene, linalool, methyl salicylate, and eugenol, with eugenol being an impurity and not a target compound.

At 20 mg L$^{-1}$, the active compounds were myrcene, linalool, methyl salicylate, and eugenol (Fig. 4A).

At concentrations of 10 mg L$^{-1}$ and below, the active compounds were linalool and methyl salicylate (Fig. 5A).

The target compounds were detected in all seven concentrations of the solution (Table 1).

## GC-EAD and GC-MS analyses of the mixed solution

GC analyses with FID were performed on the mixed solution, with a GC-EAD injection volume of 3 μL. The injection volume for GC-MS was 3 μL. Analyzed and compared the corresponding compounds in the two chromatograms, screened compounds were identified by GC-MS according to their mass spectra and retention times, in comparison with authentic standards.

At 100 mg L$^{-1}$ and 50 mg L$^{-1}$, five compounds elicited responses in female *H. armigera* antennae, and were identified as cis-3-hexen-1-ol, myrcene, linalool, methyl salicylate, and eugenol (Fig. 3B).

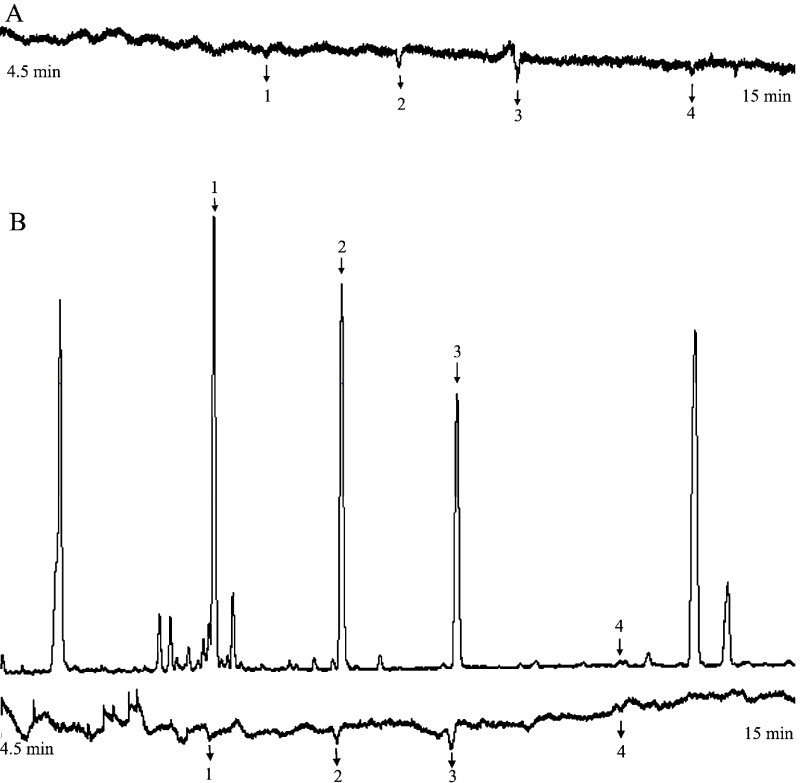

**Figure 4 The responses of female *Helicoverpa armigera* to the mixed solution (20 mg L$^{-1}$).** (A) GC-EAG-MS, (B) GC-EAD. The active compound: (1) myrcene; (2) linalool; (3) methyl salicylate; (4) eugenol.

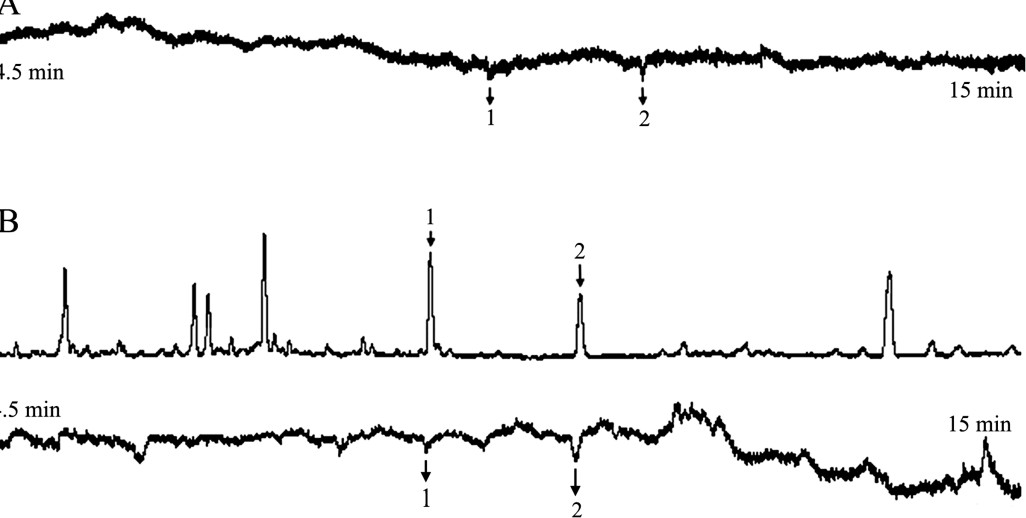

**Figure 5 The responses of female *Helicoverpa armigera* to the mixed solution (1 mg L$^{-1}$).** (A) GC-EAG-MS, (B) GC-EAD. The active compound: (1) linalool; (2) methyl salicylate.

**Table 1 The retention time of the target compounds in the mixed solution using two methods.**

| Method | Compound | Retention time (min) |
|---|---|---|
| GC-EAG-MS | Cis-3-hexen-1-ol | 5.82 |
| | Myrcene | 8.06 |
| | Linalool | 9.87 |
| | Methyl salicylate | 11.37 |
| | Trans-β-caryophyllene | 14.64 |
| GC-EAD | Cis-3-hexen-1-ol | 5.32 |
| | Myrcene | 7.34 |
| | Linalool | 9.03 |
| | Methyl salicylate | 10.54 |
| | Trans-β-caryophyllene | 13.68 |

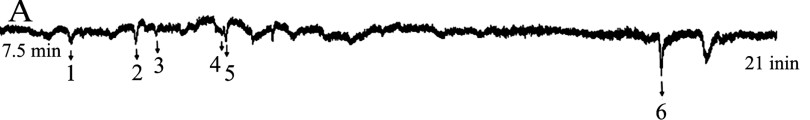

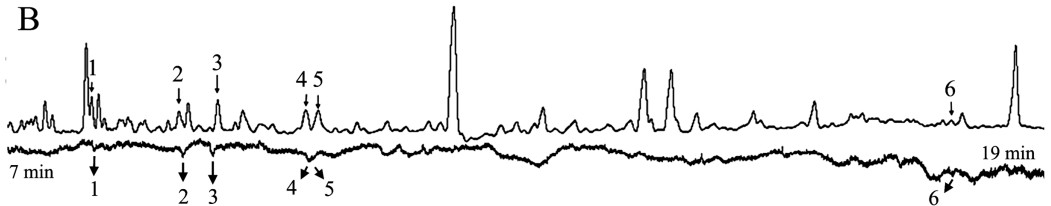

**Figure 6 The responses responses of male *Helicoverpa armigera* to volatiles from the lures.** (A) GC-EAG-MS, (B) GC-EAD. The active compound: (1) benzyl alcohol; (2) linalool; (3) 2-ethyl-p-xylene; (4) α-terpineol; (5) methyl salicylate; (6) cis-11-hexadecenal.

At 20 mg L$^{-1}$, the active compounds were myrcene, linalool, methyl salicylate, and eugenol (Fig. 4B).

At concentrations of 10 mg L$^{-1}$ and below, the active compounds were linalool and methyl salicylate (Fig. 5B).

The injection volume for GC-EAD was 1 μL. The target compounds were detected in all seven concentrations of the solution (Table 1).

## Screening of unknown compounds using two methods

### GC-EAG-MS analyses and identification of volatiles from H. armigera lures

Analyzed EAG data, six compounds consistent elicited responses in male *H. armigera* antennae (Fig. 6A). Recorded the starting time of the EAG peak of the active compounds, and found the corresponding compounds at these times in mass spectrum. These were identified by MS, according to their mass spectra and retention times, in comparison with commercial standards or synthesized samples. They were benzyl alcohol, linalool, 2-ethyl-p-xylene, α-terpineol, methyl salicylate, and cis-11-hexadecenal.

### GC-EAD and GC-MS analyses and identification of volatiles from H. armigera lures

The injection volume for GC-EAD was 3 µL. In GC analyses with FID, six compounds elicited responses in male *H. armigera* antennae (Fig. 6B). The injection volume for GC-MS was 3 µL. Analyzed and compared the corresponding compounds in the two chromatograms, screened compounds were identified by GC-MS according to their mass spectra and retention times, in comparison with commercial standards or synthesized samples. They were benzyl alcohol, linalool, 2-ethyl-p-xylene, α-terpineol, methyl salicylate, and cis-11-hexadecenal, with cis-11-hexadecenal being a known sex pheromone of *H. armigera* (*Gothilf et al., 1978*; *Piccardi et al., 1977*).

These active compounds were also detected with an injection volume of 1 µL.

## DISCUSSION

After a method is improved, it is usually compared with the original method to verify whether the improvement is accurate and feasible. The GC-EAD method has been used for decades, and it is widely recognized by researchers. The present study focuses not only on the improvement of this method but also on its verification and comparison with conventional methods.

*Weissbecker, Holighaus & Schütz (2004)* made similar improvements, but they only reported the improved method and the practical application of GC-MS/EAD. They did not perform any comparative experiments with the original method (GC-EAD). We did not know whether we would obtain the same results using GC-EAD and GC-MS/EAD under the same conditions. Therefore, in the present study, we compared the two methods —GC-EAG-MS and GC-EAD by performing single compound, known mixture, and unknown compound analyses to verify the accuracy and feasibility of the improved method. Thus, the present study is an important supplement and improvement to the work of *Weissbecker, Holighaus & Schütz (2004)*.

The method described by *Weissbecker, Holighaus & Schütz (2004)* needs to be improved in its synchronization and split ratio, and the current study has improved these aspects. *Weissbecker, Holighaus & Schütz (2004)* mentioned that due to the column and EAD interface, the delay of the EAD signal should amount to 2.5 s compared to the MS signal. The MS and the EAD signals appear synchronously using the method detailed in our manuscript. In the study conducted by *Weissbecker, Holighaus & Schütz (2004)*, the effluent of the column was split in a ratio of 1 (MS) to 1 (EAD), but in practical application, we prefer to use split ratios such as 1:2 and 1:3 (MS: EAD) because this allows as many samples as possible to enter the EAG equipment, and makes the EAG react significantly. In the present study, the split ratio in GC-EAG-MS could be switched arbitrarily, which did not affect its synchronization. We only needed to replace the column that leads to EAG. The specification of the replaced column could be calculated simply by entering the split ratio in the splitter calculator. Because we used the splitter with makeup gas (G3180B; Agilent, Santa Clara, CA, USA), it was very convenient to change the column (i.e., there was no need to empty the MS equipment). The split ratio of the GC-MS/ EAD in the studies conducted by *Weissbecker, Holighaus & Schütz (2004)* and *Paczkowski*

*et al. (2013)* was 1:1, although their method may not be able to switch the split ratio arbitrarily. If the split ratio of GC-MS/EAD can be changed, the associated columns will inevitably be changed, which will affect the delay time of the EAD signal as reported by *Weissbecker, Holighaus & Schütz (2004)*. The delay time will no longer be 2.5s and will need to be recalculated. Finally, the EAG component in our GC-EAG-MS system can be turned off such that GC-MS can operate independently, which is a highly convenient feature.

In conclusion, comparative experiments between the new and original methods (GC-EAD), which were not carried out by *Weissbecker, Holighaus & Schütz (2004)*, are reported in the current study. In addition, the accuracy and feasibility of the combined use of GC, EAG, and MS have been verified. The GC-EAG-MS method described herein is optimized and more advanced than that described by *Weissbecker, Holighaus & Schütz (2004)*.

The equipments were used for the rapid screening and identification of insect active substances in this paper. Most compounds could not elicite an antennal reaction in the EAG device at the concentration of 1 mg/L, we therefore set the minimum test concentration of the artificial solution to 1 mg L$^{-1}$. On this basis, we compared and verified between the traditional method and the improved method.

Benzaldehyde presents attractive properties to many noctuid insects (*Dötterl, Wolfe & Jürgens, 2005*), and could elicit responses in female *H. armigera* antennae(*Wang et al., 2013*). Both methods accurately screened and detected this active compound at a concentration of 1 mg L$^{-1}$. Therefore, the accuracy and sensitivity of GC-EAG-MS were the same as those of conventional methods in detecting a single compound.

Myrcene, linalool, and methyl salicylate play a role in the foraging, mating, and oviposition of *H. armigera* (*Du, 2018*; *Nebapure, 2020*). Cis-3-hexen-1-ol is a common odor of green leaves, and eugenol is a component of insect attractants (*Ladd, 1984*). These compounds also elicited responses in *H. armigera* antennae (*Chen, Zhang & Qu, 2010*; *Li et al., 2015*; *Nebapure, 2020*; *Sun et al., 2018*; *Wang et al., 2013*). The results obtained here were consistent with those of previous studies. According to a previous study, trans-β-caryophyllene played a role in prompting female *H. armigera* to lay eggs (*Hartlieb & Rembold, 1996*), and while this compound elicited responses in this species (*Cribb et al., 2007*; *Nebapure, 2020*), it did not induce a female response in the present experiment. This could be because the female moths used in this experiment were not mating, and trans-β-caryophyllene was mainly related to the oviposition of *H. armigera*. Also, the overall concentration of the solution used in this experiment was relatively low, and may not have reached the perceptual concentration to affect *H. armigera*. This result was consistent with that reported by *Chen et al. (2000)* and *Sun (2010)*.

In the mixed solution screening experiment, active compounds were accurately screened by the two methods, with consistent experimental results. Both methods detected the target compounds at the lowest concentration of the test solution, indicating that the accuracy and sensitivity of GC-EAG-MS were the same as those of conventional methods, in the detection of mixed compounds.
At 10 mg L$^{-1}$ concentration, the active compound eugenol, an impurity, did not generate an EAG response in either method but was detected by GC-EAG-MS. Under similar conditions, eugenol remained undetected by GC-EAD. The results here showed that GC-EAG-MS was superior in sensitivity compared to GC-EAD, in terms of detecting certain compounds.

The two methods used to screen the solution of unknown compounds, produced similar results by detecting the same active compounds. Subsequent behavioral experiments were excluded from the objective of this study and were therefore not addressed here.

It was a tedious process to analyze and compare two chromatograms. Such limitations occurred in previous study in our laboratory. The screening of active compounds at the ventral end of female *Holotrichia oblita* revealed that GC-MS retention time was significantly longer than that of GC-EAD, the amount of compound isolated by GC-MS was more than that of GC-EAD, and their peak shapes were different (Fig. 7). It took us a lot of time for analysis and comparison to accurately identify the active compounds. In this study, a large number of compounds were present in volatiles from *H. armigera* lures. During experimental analyses using conventional methods, differences in the retention time and peak shape of the same compound between GC-EAD and GC-MS also made the analysis and comparison of the two chromatograms a tedious process. The GC-EAG-MS method only required identifying the corresponding compound in MS, according to the time when the reaction began to appear in EAG. This eliminated the analysis and comparison step between two different chromatograms and made the whole process efficient, with a narrower margin for error.

To summarize, we used two methods to detect known and unknown compounds. By ensuring that the same sample volume reached the antennae, the experimental results of the two methods were similar, highlighting the accuracy of GC-EAG-MS. Further, ensuring that the same sample volume reached the detectors allowed the identification of target compounds at the lowest concentration of the test solution in both methods, indicating that the sensitivity of GC-EAG-MS was the same as that of conventional methods. The above results showed that the accuracy and sensitivity of the GC-EAG-MS method could be guaranteed. This method was superior to the conventional method in the detection sensitivity of some compounds, effectively reducing the possibility of missed detection. Therefore, this method could be applied to screen substances used by insects to process information.

The use of conventional methods such as GC-EAD and GC-MS to screen insect information substances inevitably used different gas chromatography. The two gas chromatograph types, carrier gases or chromatographic columns were different, so this would generate chromatographic results that were quite different in retention time, peak shape, and number of compounds for the same sample. Therefore, one of the limitations of this type of experiment was to accurately analyze and compare two chromatograms, as well as correctly screen out active substances, without making mistakes in the process. Using the GC-EAG-MS method, the two operations were combined in one system, one gas chromatograph was used throughout the experiment, which eliminated the need to compare two different chromatograms. On the premise of ensuring the accuracy of the

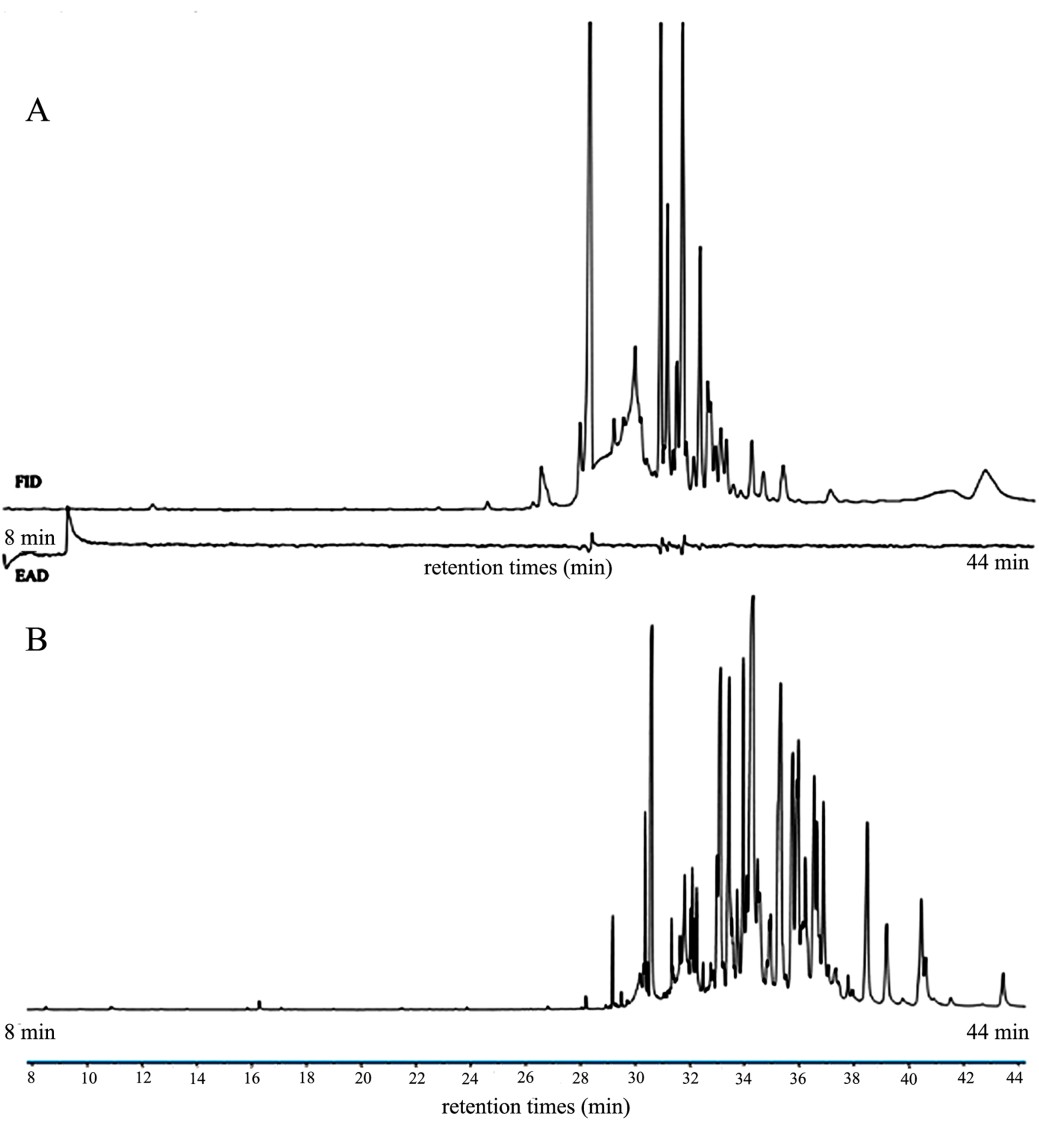

**Figure 7 GC-EAD and GC-MS chromatograms of compounds detected at the ventral end of female *Holotrichia oblita*.** (A) GC-EAD, (B) GC-MS.

experimental results, it not only saved time, but also greatly reduced the difficulty of analysis and the probability of error detection, while improving overall efficiency. Compared with the conventional method, this method eliminated the need for a GC equipment and a FID equipment, markedly reduced the experimental cost, and is economical and practical.

## CONCLUSIONS

The accuracy and sensitivity of the GC-EAG-MS method was consistent with that of the conventional method, and could replace the latter for screening active insect substances. Here, this method reduced the injection times and eliminated the need for comparison and analysis between different chromatograms, thereby notably reducing the

difficulty level of the experiment. Additionally, it was superior to the conventional method in detecting the sensitivity of some compounds. All these factors improved the efficiency of the experiment and effectively reduced error probability. Experimental costs were greatly lowered with the elimination of one GC equipment and one FID equipment, compared with conventional methods. Therefore, the GC-EAG-MS method is convenient, sensitive, economical, and practical, with wide applications in the study of plant and insect chemical ecology pending further optimization.

## ACKNOWLEDGEMENTS

Thanks to Master Guodong Kang for his help in GC-EAD experiment.

### Funding
This work was supported by the Agricultural Science and Technology Innovation Program (ASTIP-TRIC04). The funders had no role in study design, data collection and analysis, decision to publish, or preparation of the manuscript.

### Grant Disclosures
The following grant information was disclosed by the authors:
Agricultural Science and Technology Innovation Program: ASTIP-TRIC04.

### Competing Interests
The authors declare that they have no competing interests.

### Author Contributions
- Chongyang Li conceived and designed the experiments, performed the experiments, analyzed the data, prepared figures and/or tables, authored or reviewed drafts of the paper, and approved the final draft.
- Jianmin Cao performed the experiments, analyzed the data, authored or reviewed drafts of the paper, and approved the final draft.
- Xiufang Wang analyzed the data, prepared figures and/or tables, and approved the final draft.
- Pengjun Xu conceived and designed the experiments, authored or reviewed drafts of the paper, and approved the final draft.
- Xinwei Wang analyzed the data, authored or reviewed drafts of the paper, and approved the final draft.
- Guangwei Ren conceived and designed the experiments, prepared figures and/or tables, authored or reviewed drafts of the paper, and approved the final draft.

### Data Availability
The raw measurements are available in Supplementary File.

## Supplemental Information

Supplemental information for this article can be found online at http://dx.doi.org/10.7717/peerj.11510#supplemental-information.

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
