# Peer review of "Efficacy of an improved method to screen semiochemicals of insect"

_PeerJ, doi:10.7717/peerj.11510_

## Round 0.1 · original submission · Major Revisions

Please take into consideration the reviewer’s comments and provide back a point-by-point rebuttal letter addressing those concerns.

Reviewer 1 ·

Basic reporting

The authors do not seem to realize that this technique was reported almost 20 years ago, for example, see:
Weissbecker B, Holighaus G, Schütz S. Gas chromatography with mass spectrometric and electroantennographic detection: analysis of wood odorants by direct coupling of insect olfaction and mass spectrometry. J Chromatogr A. 2004 Nov 12;1056(1-2):209-16. PMID: 15595552.

Cai L, Koziel JA, O'Neal ME. Determination of characteristic odorants from Harmonia axyridis beetles using in vivo solid-phase microextraction and multidimensional gas chromatography-mass spectrometry-olfactometry. J Chromatogr A. 2007 Apr 13;1147(1):66-78. doi: 10.1016/j.chroma.2007.02.044. Epub 2007 Feb 20. PMID: 17359983.

Sebastian Paczkowski, Marta Paczkowska, Stefan Dippel, Norman Schulze, Stefan Schütz, Tilman Sauerwald, Alexander Weiß, Marco Bauer, Jörg Gottschald, Claus-Dieter Kohl,
The olfaction of a fire beetle leads to new concepts for early fire warning systems,
Sensors and Actuators B: Chemical,
Volume 183,
2013,
Pages 273-282,
ISSN 0925-4005,
https://doi.org/10.1016/j.snb.2013.03.123.

Experimental design

See comments under 1

Validity of the findings

See comments under 1

Additional comments

The coupling of GC/MS and EAD was reported almost 20 years ago, see references above. Thus, this work is not novel, and at the least, the authors must cite the previous work, and then explain how their method is different, or better, or what advantages it has over the previously published methods of coupling GC/MS and EAD

Reviewer 2 ·

Basic reporting

This MS may be is the first work that assembling gas chromatography (GC) to electroantennography (EAD), and mass spectrometry (MS) to identify semiochemicals. I think that is necessary that the authors explain the objective of the work, because there is no new information in the MS. Weissbecker et al. (2004) reported the first GC-MS/EAD with antennae of the old house borer Hylotrupes bajulus, a widespread insect pest of coniferous timbers. I think that is not correct that the authors do not include in the references of his work the paper of Weissbecker et al. (2004).The MS requires major changes, before acceptance in PeerJ. For instances, results of many of the figures do not contribute with quality of the paper, I think that they can be selected a new insects pest to demonstrate the use of GC-EAD/MS, or with the information of the fig. 2 or 3, demonstrate what is the lower amount injected of the sample to stimulate the insect antenna.

Experimental design

No comment

Validity of the findings

No comment

Additional comments

The MS requires major changes, before acceptance in PeerJ. For instances, results of many of the figures do not contribute with quality of the paper, I think that they can be selected a new insects pest to demonstrate the use of GC-EAD/MS, or with the information of the fig. 2 or 3, demonstrate what is the lower amount injected of the sample to stimulate the insect antenna.

Annotated reviews are not available for download in order to protect the identity of reviewers who chose to remain anonymous.

---

## Round 0.2 · Minor Revisions

Please take into consideration the section editor comments and provide back a point-by-point rebuttal letter addressing those concerns:

"It would be good to have an overall aim and objectives/ questions explicitly identified in the final paragraph of the introduction"

---

## Round 0.3 · accepted · Accept

Thanks for addressing the minor revisions requested. Now your manuscript is accepted in PeerJ.